# Sclerostin Alters Tumor Cell Characteristics of Oral Squamous Cell Carcinoma and May Be a Key Player in Local Bone Invasion

**DOI:** 10.3390/cells13020137

**Published:** 2024-01-11

**Authors:** Uwe Schirmer, Sina Allegra Schneider, Tatjana Khromov, Felix Bremmer, Boris Schminke, Henning Schliephake, Klaus Liefeith, Phillipp Brockmeyer

**Affiliations:** 1Institute for Bioprocessing and Analytical Measurement Techniques, D-37308 Heiligenstadt, Germany; uwe.schirmer@iba-heiligenstadt.de (U.S.); allegra.schneider@alumni.fh-aachen.de (S.A.S.); klaus.liefeith@iba-heiligenstadt.de (K.L.); 2Department of Clinical Chemistry, University Medical Center Goettingen, D-37075 Goettingen, Germany; tatjana.khromov@med.uni-goettingen.de; 3Institute of Pathology, University Medical Center Goettingen, D-37075 Goettingen, Germany; felix.bremmer@med.uni-goettingen.de; 4Department of Oral and Maxillofacial Surgery, University Medical Center Goettingen, D-37075 Goettingen, Germany; boris.schminke@med.uni-goettingen.de (B.S.); schliephake.henning@med.uni-goettingen.de (H.S.)

**Keywords:** oral squamous cell carcinoma, bone invasion, sclerostin, SOST

## Abstract

Localized jawbone invasion is a milestone in the progression of oral squamous cell carcinoma (OSCC). The factors that promote this process are not well understood. Sclerostin is known to be involved in bone metabolism and there are preliminary reports of its involvement in bone tumors and bone metastasis. To identify a possible involvement of sclerostin in the bone invasion process of OSCC, sclerostin expression was analyzed in vitro in two different human OSCC tumor cell lines by quantitative real-time polymerase chain reaction (qRT-PCR), and the effect of recombinant human (rh)-sclerostin treatment on tumor cell capabilities was evaluated using proliferation, migration, and invasion assays. Undifferentiated human mesenchymal stem cells (hMSCs) were osteogenically differentiated and co-cultured with OSCC tumor cells to demonstrate potential interactions and migration characteristics. Sclerostin expression was evaluated in clinical cases by immunohistochemistry at the OSCC–jawbone interface in a cohort of 15 patients. Sclerostin expression was detected in both OSCC tumor cell lines in vitro and was also detected at the OSCC–jawbone interface in clinical cases. Tumor cell proliferation rate, migration and invasion ability were increased by rh-sclerostin treatment. The migration rate of tumor cells co-cultured with osteogenically differentiated hMSCs was increased. The results presented are the first data suggesting a possible involvement of sclerostin in the bone invasion process of OSCC, which deserves further investigation and may be a potential approach for drug-based tumor therapy.

## 1. Introduction

Oral squamous cell carcinoma (OSCC) is one of the most common forms of malignant head and neck tumors, with worldwide impact [1]. In advanced tumor stages, it is associated with poor patient prognosis and reduced health-related quality of life (HRQOL) [2,3,4]. Invasion of the locally adjacent jawbone occurs in 12% to 56% of cases [5,6]. It has been shown that OSCC bone invasion is associated with the incidence of lymph node metastasis and a poor prognosis [7,8,9].

Destroyed bone sections require resection and reconstruction with free vascularized bone grafts, primarily from the fibula [10], scapula [11] or iliac crest [12]. Such procedures are time-consuming, surgically complex, and associated with functional and aesthetic limitations and patient morbidity [13]. OSCC bone invasion is a complex process involving a variety of different cellular and molecular mechanisms that are still poorly understood. However, a better understanding of these processes could have a significant impact on patient care.

The Wnt (wingless/integrated) signaling pathway is a highly conserved pathway in biological evolution that is involved in cell proliferation, differentiation, migration and polarity of physiological processes such as embryonic development and organ morphology, but also in the pathology of various diseases [14]. It has also been shown to play an important role in the regulation of bone metabolism and osteoblastic gene expression [15,16], as well as in the initiation and development of various malignant tumors. For example, it has been implicated in the malignant transformation of intestinal epithelia, hyperplasia of breast tissue, skin cancer, and cell proliferation in lung cancer [17,18]. It has been also associated with inhibition of breast cancer cell invasion and downregulation of four-and-a-half LIM domain protein 2 (FHL2) in human osteosarcoma cells [19,20]. The pathway is involved in the regulation of chondrocyte proliferation and mesenchymal stem cell fate [16,21]. Inappropriate stimulation of the Wnt signaling pathway has been shown to promote several pathological conditions, including malignancies [22].

The SOST gene (sclerostin gene) encodes sclerostin, a protein that acts as a negative regulator of the Wnt signaling pathway [23]. By binding to low-density lipoprotein receptor protein 5 and 6 (LRP5/6) on the surface of osteoblasts, it reduces bone formation and promotes bone resorption [23]. In addition, it competes with type I and type II bone morphogenetic protein (BMP) receptors for binding to BMPs, thereby reducing BMP signaling and suppressing mineralization of osteoblastic cells [23,24]. Sclerostin has been implicated in the pathogenesis of several Wnt-related musculoskeletal disorders [25]. There is evidence that sclerostin is associated with the biology of bone metastases and primary bone tumors in several entities. For example, breast cancer often causes bone metastases and osteolytic bone destruction by stimulating osteoclasts to resorb bone and preventing osteoblasts from forming new bone [26]. Sclerostin is overexpressed in breast cancer tumor tissue and cells and promotes growth, invasion, and bone osteolysis [27]. Inhibition of sclerostin reduces migration and invasion of this tumor entity in a time- and dose-dependent manner [27]. Runt-related transcription factor 2 (Runx2) is known to mediate activation of osteoclast activity and inhibition of osteoblast differentiation by metastatic breast cancer cells [26]. Runx2 requires co-activator core-binding factor beta (CBFb) to regulate gene expression in breast cancer cells. The combination of Runx2 and CBFb mediates the inhibition of osteoblast differentiation through the induction of sclerostin [26]. In vivo models in mice with breast cancer-related bone metastases have shown that pharmacological sclerostin inhibition reduces metastatic burden [28], prolongs animal survival [27] and prevents cancer-related bone destruction [27,29].

A similar effect as in breast cancer has also been demonstrated in multiple myeloma. Myeloma cells are known to produce and shed sclerostin into the serum/plasma of patients [30]. Increased serum sclerostin levels correlate with more extensive bone disease and negative myeloma features [30]. Pharmacological sclerostin inhibition prevented bone loss and preserved bone strength in preclinical studies without significantly affecting tumor growth [31].

Several risk factors and causative genes for the development of osteosarcoma have been reported in the literature [28]. However, the etiology remains largely unknown. Bone formation is a common phenomenon in all types of osteosarcoma. Sclerostin has also been shown to suppress the proliferative and migratory capacity of osteosarcoma cells, and administration of sclerostin inhibits tumor growth in mice and prolongs animal survival [28].

In prostate cancer, sclerostin expression is reduced and can be used in combination with BMP-6 and noggin expression as a prognostic factor for metastatic progression [30].

The involvement of sclerostin in the bone invasion by OSCC has not yet been demonstrated. In a pilot study, we have shown that a human OSCC tumor cell line treated with transforming growth factor beta (TGF-β) upregulates SOST gene expression and that sclerostin protein expression has a significant prognostic impact on patients [32]. To further explore these interesting findings, we investigated the influence of sclerostin on the proliferation, migration, and invasion of OSCC tumor cells in an in vitro approach and evaluated sclerostin expression at the OSCC–jawbone interface in clinical cases. We show here for the first time that sclerostin can alter the properties of tumor cells toward a more aggressive phenotype and contribute to a tumor-friendly microenvironment at the OSCC–jawbone interface, thereby promoting bone invasion. These findings may contribute to a better understanding of bone invasion in OSCC and, with therapeutic sclerostin inhibition, have the potential to contribute to broader clinical relevance by reducing locally destroyed bone sections.

## 2. Materials and Methods

In this study, a possible influence of sclerostin on the bone invasiveness of OSCC was evaluated using three different approaches: (1) treatment of OSCC tumor cells with rh-sclerostin and its influence on cellular characteristics proliferation, migration and invasion; (2) co-cultivation of OSCC tumor cells with osteogenically differentiated human bone marrow-derived human mesenchymal stem cells (hMSCs) and its influence on tumor cell migration; and (3) immunohistochemical analysis of the OSCC–jawbone interface in human tissue samples.

### 2.1. Cell Culture

PCI-13 (UPCI, Pittsburgh, PA, USA) [33] and UPCI-SCC-040 (SCC-040) cells (DSMZ, Braunschweig, Germany) were cultured in DMEM + GlutaMAX (31966021, Thermo Fisher, Waltham, MA, USA) and MEM Earle’s (FG0325, Merck, Darmstadt, Germany) with 10% fetal bovine serum (*v*/*v*) and 1% penicillin–streptomycin (*v*/*v*), respectively (37 °C, 5% CO_2_, 80% humidity). For cell expansion or cell seeding, cells were enzymatically detached from the surface using 0.05% (*w*/*v*) trypsin–0.02% (*w*/*v*) EDTA and resuspended in serum-free medium.

Human bone marrow-derived mesenchymal stem cells (hMSCs) were provided by the Bader laboratory of the Centre for Biotechnology and Biomedicine, Leipzig, Germany. The isolation of hMSCs from donors was approved by the local ethics committee (Saxony Regional Authority, EK-BR-86/14-1). HMSCs were cultured in DMEM (P04-01159, PAN-Biotech, Aidenbach, Germany) containing 10% fetal bovine serum (*v*/*v*), 0.2% gentamicin/ampicillin (*v*/*v*) and 4 mM glutamine (37 °C, 5% CO_2_, 80% humidity). For differentiation, further supplements were added to the medium (2 mM glutamine; 0.05 µM sodium ascorbate; 0.1 dexamethasone; 10 mM β-glycerol phosphate; 0.015 mM CaCl_2_; 0.05 µM vitamin D3). Before each experiment, cells were stained with trypan blue and counted using a cell counter (Logos Biosystems, Anyang, Republic of Korea). All cell biological experiments were performed in triplicate in at least three separate experiments.

### 2.2. qRT-PCR Analysis

For qRT-PCR analysis, RNA was isolated from cells using a Qiagen RNA Isolation Kit (74104, Qiagen, Hilden, Germany) according to the manufacturer’s instructions. Prior to RNA isolation, the analyzed cells were seeded in a 6-well plate and grown to 80% confluence. For the analysis of differentiated hMSCs, cells were cultured as described above and RNA was isolated after 14 days of cultivation with differentiation media.

The isolated RNA was used as input for subsequent cDNA synthesis using RevertAid first-strand cDNA synthesis from Thermo Fisher (K1622, Thermo Fisher, Waltham, MA, USA). From each sample, 1 µg of RNA was used for reverse transcription. Finally, 10 ng of cDNA was used for each SYBR green qRT-PCR reaction. A 25 µL reaction volume was prepared for each well of a 96-well plate using Luna^®^ Universal Probe qPCR Master Mix (New England Biolabs, Ipswich, MA, USA) according to the manufacturer’s instructions. The Ct value was determined in a 40-cycle reaction using a QuantStudio 3 (Thermo Fisher, Waltham, MA, USA). Delta Ct was further calculated using the design and analysis software DA2 version 2.6.0 from Thermo Fisher (Thermo Fisher, Waltham, MA, USA) using the relative quantification method. GAPDH was used for endogenous normalization, and Ct values of untreated hMSCs were used as reference. Desalted oligonucleotides were purchased from Merck (Merck, Darmstadt, Germany). The sequences of all primer pairs are shown in Table 1.

### 2.3. Proliferation Assay

Cell proliferation was measured using the RealTime GloTM assay (G9711, Promega, Madison, WI, USA). The assay was performed according to the manufacturer’s instructions and the luciferase reaction was measured every 24 h. To analyze the influence of recombinant human (rh)-sclerostin (100-49, Peprotech, Hamburg, Germany) on cell proliferation, 1500 cells were seeded per well of a 96-well plate and four different concentrations of sclerostin (0; 1; 5; 10 ng/mL) were tested. To achieve a critical number of technical replicates in each independent experiment, six wells were seeded per concentration. After initial seeding, proliferation was measured every 24 h. The medium was completely changed before each measurement and new sclerostin was added with each medium change. Reduction in luciferase substrate was measured using a microtiter plate reader (Biotek, Winooski, VT, USA). The proliferation assay was repeated three times and all experiments showed the same trend. However, due to the large variance, statistical analysis over all three experiments was not sufficient. Therefore, a representative data set is shown in the results.

### 2.4. Migration and Invasion Assays

The SOST gene has previously been shown to influence the cellular phenotype of tumor cells [27]. There is clear evidence that neutralizing antibody treatment reduces the migration and invasion of breast tumor cells. Since OSCC is a highly invasive tumor in bone tissue where sclerostin is endogenously expressed, it was hypothesized that OSCC cells might also be affected by higher levels of SOST in the environment. Therefore, whether OSCC cell lines show the expected phenotype regarding migration and invasion when SOST is applied in a gradient was tested.

Migration and invasion were performed as described previously [34] with some modifications. Briefly, for migration and invasion, 20,000 PCI-13 cells were seeded in the upper volume of a transwell insert with a pore size of 8.0 µm for a 12-well plate (9318012 cellQART^®^, Northeim, Germany, Figure 1). For invasion analysis, the membrane of each transwell insert was additionally coated with 1 mg/mL collagen type 1 solution (50301, Matrix Bioscience, Moerlenbach, Germany). To avoid collagen fibrillation, the collagen was diluted in ice-cold 0.1% acetic acid and 200 µL of the solution was added to each membrane. All membranes were then incubated for a further 2 h under cell culture conditions. Excess collagen was again removed and 1 mL of cell culture medium was added to the transwell to neutralize the membrane-bound collagen. The inserts were then incubated for a further 2 h under cell culture conditions. The cells were diluted in 400 µL of serum-free cell culture medium. The lower compartment was filled with 1 mL of cell culture medium containing 10% fetal calf serum. In addition, the medium in the lower compartment was supplemented with sclerostin. To investigate the sensitivity of the cells to sclerostin, three different concentrations (0, 1, 5 ng/mL) were prepared for each experiment. The wells without SOST in the lower volume were also the control, as only reduced migration/invasion was expected in these wells compared to the wells with SOST in the medium. After seeding, the cells were incubated for 24 h under cell culture conditions. The cells were then fixed on the membrane with 4% PFA for 20 min at room temperature. Before and after fixation, the membranes were washed three times by adding 500 µL in PBS in the upper and lower compartments of the transwell. Finally, the cells were stained with 5 µg/mL Hoechst 33258 (K1622, Thermo Fisher, Waltham, MA, USA) and washed three times with PBS. After staining, cells on the upper side of the membrane were removed from the insert using an ear swab. Migrated/invaded cells on the underside of the membrane were imaged with a fluorescence microscope at 10× magnification (AxioImager 2, Zeiss, Oberkochen, Germany). An area of at least 5 × 5 mm was imaged from each membrane. Automated cell counting was performed using Arivis4D version 3.2 (Zeiss, Oberkochen, Germany). To avoid side effects and high background signal in the outer areas of the membrane, a standardized region of interest was defined and positioned in the central area of each stitched image. Finally, stained nuclei were counted using a particle detection tool of the Arivis4D software (version 3.2). Each independent experiment was repeated at least three times with three technical replicates for each condition.

### 2.5. Co-Cultivation Experiments

Co-cultivation was performed as described above with slight modifications regarding the cultivation of hMSCs in the lower compartment of the transwell. For the differentiation of hMSCs, 31,500 cells were seeded in a 12-well plate from cells with passages fewer than 9. After three days of cultivation with proliferation medium, the cells were further incubated with differentiation medium until day 14. The medium was changed every three days, the last time two days before the start of co-culture. After this period, the transwell was placed in the well and PCI-13 cells were seeded at the same density in the upper compartment of the transwell as described above. In the experimental design, four different conditions were analyzed. Two conditions with cells and two conditions without cells. The PCI-13 cells were incubated with pre-differentiated and undifferentiated hMSC and the respective hMSC cell culture media only in the lower compartment. Co-incubation with hMSCs was performed to show that hMSCs are able to induce OSCC cell migration, and it was expected that the effect would correlate with differentiation due to the increasing SOST level during MSC differentiation. The samples without cells were chosen as controls to exclude side effects due to the different media composition. The co-culture was incubated for 24 h and analyzed as described above. The experiment was repeated three times with three technical replicates for each condition in a single experiment.

### 2.6. Scratch Assay

For the scratch assay, a confluent cell monolayer was established by seeding 47,500 PCI-13 cells in a well of a 24-well plate and incubating for 24 h under cell culture conditions. The monolayer was scraped by hand with a 2.5 mL pipette tip to set the scratch. Immediately after scraping, a first image was taken at 4× magnification (Olympus IX81, EVIDENT, Hamburg, Germany). Further images were taken every 12 h over a period of 36 h. To test the effect of sclerostin on the cell phenotype, cells were incubated in media containing three different concentrations of sclerostin (0, 1 and 5 ng/mL). Scratch size was analyzed using Fiji Image J version 2.14.0/1.54f (https://fiji.sc/, accessed on 15 October 2023) and the cell-free area was measured using the wound healing size tool. Before using the tool, a region of interest (ROI) of a defined size was placed in the center of the image for all images, and only the area within the ROI was finally analyzed. All experiments were repeated three times with three technical replicates for each condition in a single experiment.

### 2.7. Semi-Quantitative Immunohistochemical Evaluation of Sclerostin Expression at the OSCC–Jawbone Interface in Clinical Cases

Tissue samples from 15 OSCC patients undergoing primary surgical treatment between 2016 and 2020 were retrospectively used for visualization of the OSCC–jawbone interface and immunohistochemical sclerostin evaluation. The clinical characteristics of the patients are given in Table 2. Patients provided written informed consent before participation in the study. The study was approved by the clinical ethics committee of the University Medical Center Goettingen (vote 07/06/09, updated April 2018).

Tissue samples were collected immediately after tumor resection, fixed in neutral buffered 4% formalin, and embedded in paraffin. Immunohistochemical reactions (Table 3) were performed on 2 μm sections using a fully automated slide stainer (Agilent Technologies, Santa Clara, CA, USA). Tissue slides were digitized using a Motic EasyScan One slide scanner (Motic, Hong Kong, China) at 80× magnification and 0.13 μm/pixel resolution. For semi-automated, semi-quantitative immunohistochemical analysis, we used the open-source image analysis software quPath (version 0.4.4, https://qupath.github.io/, accessed on 1 November 2023) in the default settings [35].

In each case, three different regions of interest (ROIs) were digitally defined at the OSCC–jawbone interface. Each ROI was approximately 1 cm^2^ in size. As part of the immunohistochemical evaluation, color separation was achieved using color deconvolution with spot vector and background data [36]. First, automatic cell detection was performed using default software settings to identify all cells in each ROI. This function produces up to 33 individual measurements, which were refined using the QuPath “add smoothed features” command. A two-way random tree classifier was then trained to distinguish OSCC tumor cells from other cell or tissue types. Intensity thresholds were defined in the software’s default settings to further subdivide tumor cells with negative, weak, moderate, or strong positive staining based on mean optical DAB densities [35].

For each ROI, a histoscore (H-score) was calculated by adding 3×% strongly stained tumor cells, 2×% moderately stained tumor cells, and 1×% weakly stained tumor cells [37], resulting in scores ranging from 0 (all tumor cells negative) to 300 (all tumor cells strongly positive). The means of the three different ROIs were calculated for statistical analysis.

### 2.8. Statistical Analysis

All data were tested for normal distribution using the Shapiro–Wilk test. Since all data were normally distributed, group comparisons were performed using one-way ANOVA analysis. Post hoc comparisons were performed using Tukey’s tests. All statistical analyses were performed at a significance level of α = 5% using Prism 10.10 software (GraphPad, La Jolla, CA, USA). A *p*-value less than 0.05 was considered statistically significant.

## 3. Results

### 3.1. Sclerostin Expression in Different OSCC Tumor Cell Lines

To detect sclerostin expression in the two different OSCC tumor cell lines (SCC-040 and PCI-13), qRT-PCR analysis was performed and cycle threshold (CT) values were determined as shown in Figure 2. Sclerostin expression was detected in both cell lines, although its expression was higher in the OSCC cell line SCC-040 (MV 24.57) than in the PCI-13 cell line (MV 31.26).

### 3.2. Evidence of Osteogenic Differentiation of hMSCs and Their Sclerostin Expression

For co-culture experiments, hMSCs were cultured in osteogenic differentiation medium (DMEM) for 14 days. To demonstrate successful differentiation, qRT-PCR analysis of known osteogenic markers (ALPL, COL1A1, osteocalcin, osteopontin, osterix, and RUNX2) [38] and sclerostin expression was performed. Undifferentiated hMSCs were used as a control and GAPDH was used as endogenous reference gene. The analysis revealed overexpression of the known osteogenic markers ALPL, osteocalcin, and osterix, indicating successful differentiation of the hMSCs. In addition, higher sclerostin expression was detected in the osteogenically differentiated hMSCs compared to the undifferentiated controls (Figure 3).

### 3.3. Effect of Sclerostin Treatment on Tumor Cell Proliferation

To evaluate the influence of sclerostin on tumor cell proliferation, the two OSCC tumor cell lines SCC-040 and PCI-13 were treated with different concentrations (1 ng/mL, 5 ng/mL, and 10 ng/mL) of rh-sclerostin for different time periods (24 h, 36 h, 48 h, 60 h, and 72 h). The experiments showed a significant increase in tumor cell proliferation for the SCC-040 tumor cell line at the higher sclerostin concentrations (5 and 10 ng/mL) after all treatment times (24 h, 36 h, 48 h, 60 h and 72 h, all *p*-values < 0.05). In addition, a significant effect was also demonstrated for the highly proliferative PCI-13 tumor cell line for all sclerostin concentrations after treatment periods of 24 h, 36 h and 48 h (all *p*-values < 0.05, Figure 4).

### 3.4. Effect of Sclerostin Treatment on Tumor Cell Migration and Invasion

A Boyden chamber assay was performed to evaluate the influence of sclerostin on tumor cell migration and invasion. Preliminary experiments have shown that the low proliferative SCC-040 tumor cell line grows strongly in clusters, and evaluation of migration and invasion using the described assay was not practical. For this reason, migration and invasion assays were performed only on the PCI-13 cell line. For this purpose, PCI-13 cells were placed in the upper compartment of Boyden chambers and treated with different concentrations of sclerostin (1 and 5 ng/mL). After a period of 24 h, the migration of the tumor cells to the lower compartment of the chamber was evaluated by using fluorescence microscopy and cell nucleus counting with Arivis4D.

The analysis revealed significantly increased PCI-13 tumor cell migration for the higher concentration of sclerostin (5 ng/mL) compared to the low concentration (1 ng/mL) and the untreated controls (0 ng/mL), as shown in Figure 5a (all *p*-values < 0.05). Subsequently, the porous membrane of the Boyden chamber assay was additionally coated with collagen to simulate tumor cell invasion into the extracellular matrix. The tumor cells were treated with 1 and 5 ng/mL rh-sclerostin, equivalent to the migration assay, and tumor cell invasion into the membrane was evaluated by fluorescence microscopy. The results showed increased tumor cell invasiveness with increasing sclerostin concentration, with only the higher dose of sclerostin (5 ng/mL) showing a statistically significant result (*p* < 0.05), as shown in Figure 5b.

### 3.5. Migration Analysis by Scratch Assay

To additionally evaluate PCI-13 cell migration, a scratch assay was performed under sclerostin treatment of tumor cells at different concentrations (1, 5 ng/mL) and time periods (0 h, 12 h, 24 h, 36 h). As shown in Figure 6 and Figure 7, scratch closure was observed regardless of the sclerostin concentration.

### 3.6. Tumor Cell Migration in Co-Culture with Osteogenic Differentiated hMSC Cells

To simulate the influence of osteogenic active cells on tumor cell characteristics, PCI-13 cells were co-cultured with osteogenically differentiated and undifferentiated hMSCs and the tumor cell migration rate was evaluated equivalently to that described in the previous experiments. In addition, tumor cells were treated with differentiation medium alone. Monocultured PCI-13 in normal cell culture medium served as a control. The study revealed the highest tumor cell migration rate for the PCI-13 cells co-cultured with osteogenically differentiated hMSCs (*p*-value < 0.05), as shown in Figure 8. Lower tumor cell migration rates were observed both in co-culture with undifferentiated hMSCs and in monoculture with the differentiation medium, but still higher than the untreated controls.

### 3.7. Sclerostin Expression of Tumor Cells at the OSCC–Jawbone Interface

To evaluate sclerostin expression in tumor cells at the OSCC–jawbone interface in clinical cases, histological slides from 15 patients with bone-invasive OSCC were prepared and immunohistochemically evaluated. H-score values ranged from 61 to 127 (MV 94, SD 21), indicating significant sclerostin expression in tumor cells at the OSCC–jawbone interface. The results of all H-score values are shown in Table 1. H-score values range from 0 to 300, with 0 indicating that no cells are positive and 300 indicating that all cells are strongly positive. The OSCC–jawbone interface is highlighted in Figure 9, indicating significant sclerostin expression in tumor cells and bone-associated osteoclasts.

## 4. Discussion

The process of local jawbone invasion in OSCC is a milestone in tumor progression [9], indicating aggressive tumor biology [39], and is considered a prognostic indicator associated with increased recurrence frequency and decreased survival rates [39,40,41,42]. The negative clinical impact in terms of impaired functionality, aesthetics and associated reduced HRQOL [43] is a major challenge that requires aggressive oncological treatment with resection of the involved bone sections and functional and aesthetic plastic reconstruction [44]. Its clinical significance has led to the inclusion of bone invasion status in the T-status of the International Union Against Cancer (UICC) and American Joint Committee on Cancer (AJCC) OSCC staging systems, reflecting its importance in disease staging and treatment decision-making [45]. However, the mechanisms of OSCC bone invasion involve complex cellular and molecular processes, including regulation of osteoclast differentiation [46], bone resorption [9], and expression of various molecular targets such as parathyroid hormone-related protein [47], fractalkine [48], and the Axin2–snail axis [49], and remain poorly understood [9].

The invasion of solid tumors into bone is known to be based on a complex interaction between tumor cells, bone-forming osteoblasts, and bone-resolving osteoclasts [50]. Tumor-derived cytokines in the tumor microenvironment (TME) alter the balance of osteoclast and osteoblast activity, disrupting physiological bone homeostasis and promoting bone destruction by enhancing osteoclast and inhibiting osteoblast function [50,51,52]. Osteolytic bone lesions provide niches for tumor cells to further interact with osteoblasts and osteoclasts, creating a vicious cycle that perpetuates tumor growth in bone [52,53]. While the interaction between tumor cells and osteoclasts is well established, the influence of tumors on osteoblasts at sites of bone invasion has received less attention [54,55].

There is increasing evidence in the literature that sclerostin plays a key role in several Wnt-related musculoskeletal disorders [25,31], primary bone tumors of various entities [28,30], and the development of bone metastases [26,27,28,29]. Mendoza-Villanueva and colleagues have shown that Runx2 and CBFb inhibit osteoblast differentiation in bone metastatic breast cancer cells and that this inhibition is mediated by the induction of sclerostin expression [26]. In addition, two target genes, IL-11 and GM-CSF, were identified as being involved in the enhanced osteoclast activation [26]. Wijenayaka and colleagues treated human pre-osteocyte cultures and mouse osteocyte-like cells with recombinant sclerostin and observed that it upregulated the expression of receptor activator of nuclear factor kappa B (RANKL) mRNA and downregulated osteoprotegerin (OPG) mRNA, leading to increased osteoclast activity [56]. In addition, sclerostin reduces BMP signaling and suppresses osteoblast cell mineralization [24]. In breast cancer, the SOST gene was found to interact with signal transducer and activator of transcription 3 (STAT3) and enhance TGF-β/KRAS (Kirsten rat sarcoma virus) signaling, leading to increased tumor growth and bone metastasis [57]. Knockdown of SOST has been shown to activate the Wnt/β-catenin signaling pathway to promote proliferation and invasion and decrease apoptosis in retinoblastoma cells [58].

In the present investigation, we demonstrate significant levels of sclerostin expression in tumor cells at the OSCC–jawbone interface in patients with locally advanced, bone-invasive, growing OSCC. OSCC tumor cells can actively synthesize sclerostin in vitro, and their cellular properties in terms of proliferation, migration, and invasion can be enhanced by sclerostin treatment, which appears to be dose- and time-dependent, with these effects most evident at higher doses of sclerostin and longer treatment times. Initial data from our co-culture experiments with osteogenically differentiated hMSCs confirm this effect. Similar results have been reported by Zhu and colleagues [27]. The effect of sclerostin on the proliferation, migration, and invasion abilities of two different breast cancer cell lines (MCF-7 and MDA-MB-231) was investigated by inhibiting sclerostin with an inhibitory antibody at different doses (1 and 4 µg/mL) and time periods (1, 2 and 3 days). While no effect on tumor cell proliferation was observed, migration and invasion rates were significantly reduced. The authors concluded that sclerostin inhibition may reduce the potential of breast cancer tumor cells to form bone metastases [27].

To date, no studies have been published demonstrating a direct relationship between sclerostin function and bone invasion in OSCC. The data presented here provide the first evidence of such an association, which is of high clinical interest. We have recently shown that human OSCC tumor cells upregulate sclerostin expression under TGF-β treatment and that sclerostin expression has significant prognostic implications for patients [32]. Tumor cells that increase their migratory and invasive capabilities undergo an epithelial–mesenchymal transition (EMT) process, and the TGF-β signaling pathway is known to be primarily involved [59]. The increase in sclerostin expression during this process may represent a priming of tumor cells prior to local bone invasion. It may play an important role in modulating tumor cell behavior by affecting key signaling pathways such as Wnt/β-catenin and TGF-β/KRAS, which ultimately influences the proliferation, migration and invasion of OSCC tumor cells. In addition, sclerostin appears to be involved in the establishment of a tumor-friendly microenvironment by inhibiting osteoblast differentiation and function and promoting osteoclast formation and activity at the site of tumor invasion [26,56].

However, the results of this study must be interpreted with caution because the functional studies were performed on only two OSCC tumor cell lines, and OSCC belongs to a heterogeneous group of tumors with different tumor cell clones and characteristics. Further studies are needed to evaluate these effects on different stages of OSCC transformation and to focus on the molecular background and possible interaction partners. The clinical presentation of sclerostin expression at the jawbone interface of OSCC was demonstrated in only a small number of cases, with no statistically significant effect on clinical parameters, such as factors of the TNM classification system. Future studies need to overcome this limitation and include osteoblast and osteoclast activity in this region.

As anti-sclerostin drugs such as romosozumab [60] and raloxifene [61] are already approved for the treatment of overt postmenopausal osteoporosis, the impact of these drugs on sclerostin–bone metabolism interactions in OSCC should also be evaluated to assess a potential therapeutic benefit for the treatment of patients with advanced tumor stage. Inhibition of sclerostin may offer the clinical potential to reduce local bone destruction in OSCC, reduce the extent of surgical intervention with corresponding functional and aesthetic benefits, and improve patient quality of life.

## 5. Conclusions

The data presented here provide a first indication of a possible involvement of sclerostin function in OSCC bone invasion by altering the cellular properties of tumor cells towards a more aggressive phenotype. The immunohistochemical data confirm high sclerostin expression at the OSCC–jawbone interface, which may maintain a tumor-friendly microenvironment and support bone invasion. Further functional and clinical studies are needed to elucidate the molecular background and to potentially exploit drug inhibition of sclerostin therapeutically to reduce local jawbone destruction and improve patient quality of life.

## Figures and Tables

**Figure 1 cells-13-00137-f001:**
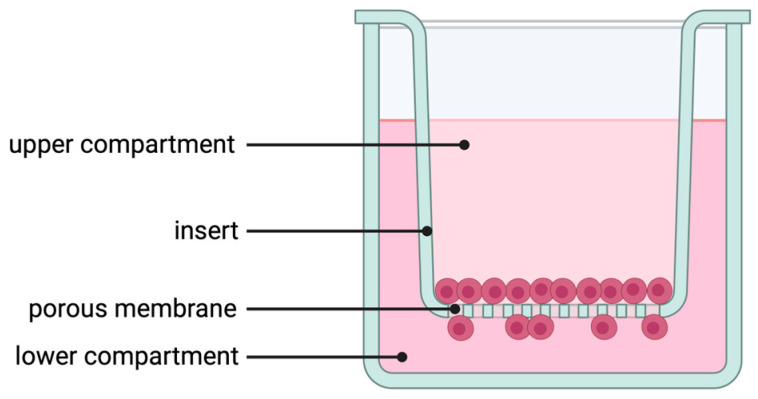
Illustration of the Boyden chamber assay used for migration, invasion, and co-cultivation experiments. For the migration assay, PCI-13 OSCC tumor cells were added to the upper compartment of the transwell insert and cell passage was determined at 8 µm pore size. For invasion analysis, the membrane of each transwell insert was additionally coated with 1 mg/mL collagen type 1 solution. For co-culture, PCI-13 tumor cells were cultured in the upper compartment and hMSCs were cultured in the lower compartment of the transwell.

**Figure 2 cells-13-00137-f002:**
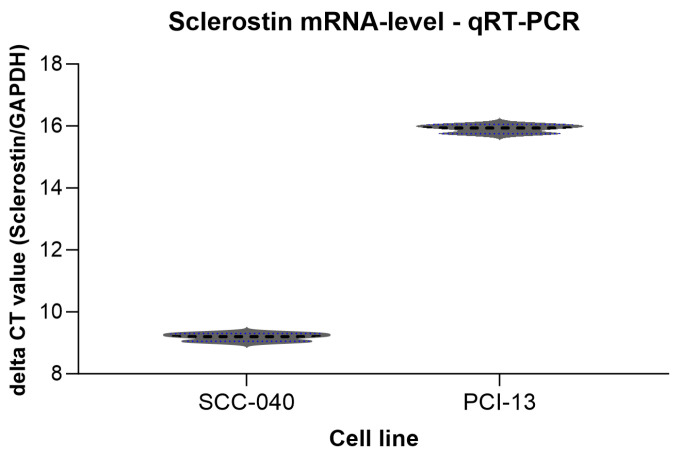
Graphical representation of the averaged delta CT values from the qRT-PCR experiments for the two different OSCC tumor cell lines (SCC-040 and PCI-13). Higher CT values indicate low sclerostin expression, while lower values indicate higher expression. For normalization, the delta CT value was calculated with the CT values of GAPDH and sclerostin (SOST) from each cell line. SCC-040 tumor cells have higher sclerostin expression than PCI-13 tumor cells.

**Figure 3 cells-13-00137-f003:**
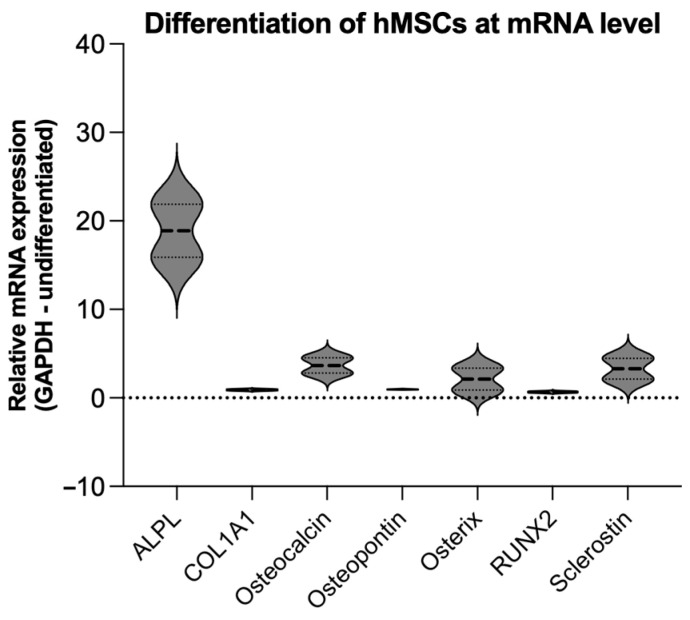
Graphical representation of the relative mRNA expression from the qRT-PCR experiments for the different osteogenesis markers ALPL, COL1A1, osteocalcin, osteopontin, osterix, and Runx2, as well as sclerostin expression. Undifferentiated hMSCs served as controls, GAPDH was used as internal housekeeping gene. Successful osteogenic differentiation of hMSCs is shown.

**Figure 4 cells-13-00137-f004:**
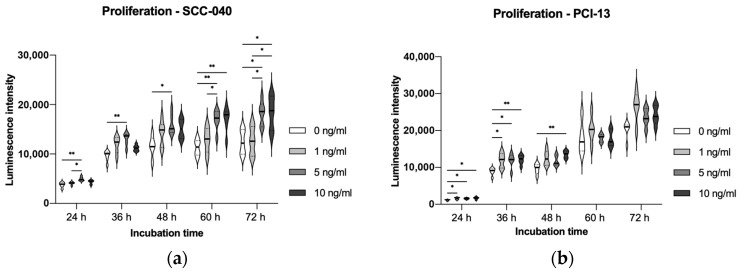
Illustration of tumor cell proliferation under sclerostin treatment at different concentrations (1, 5, and 10 ng/mL) and time periods (24 h, 36 h, 48 h, 60 h, and 72 h). Untreated tumor cells (0 ng/mL) were used as controls: (**a**) SCC-040 cell line; (**b**) PCI-13 cell line. Statistical test: one-way ANOVA with Tukey’s tests for post hoc comparisons; * *p*-values between 0.01 and 0.05 (significant); ** *p*-values between 0.001 and 0.01 (very significant). A significant influence of sclerostin on the tumor cell proliferation rate is shown, which seems to be time- and dose-dependent.

**Figure 5 cells-13-00137-f005:**
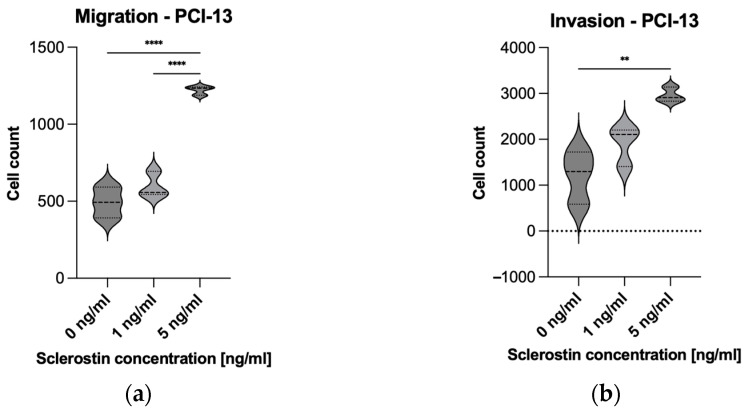
Illustration of PCI-13 tumor cell migration and invasion after 24 h under sclerostin treatment at different concentrations (1 and 5 ng/mL). Untreated tumor cells (0 ng/mL) were used as controls: (**a**) Migration of PCI-13 tumor cells; (**b**) Invasion of PCI-13 tumor cells. Statistical test: one-way ANOVA with Tukey’s tests for post hoc comparisons; ** *p*-values between 0.001 and 0.01 (very significant); **** *p*-values < 0.0001 (extremely significant). A significant effect of sclerostin on tumor cell migration and invasion rate is shown, which seems to be dose-dependent.

**Figure 6 cells-13-00137-f006:**
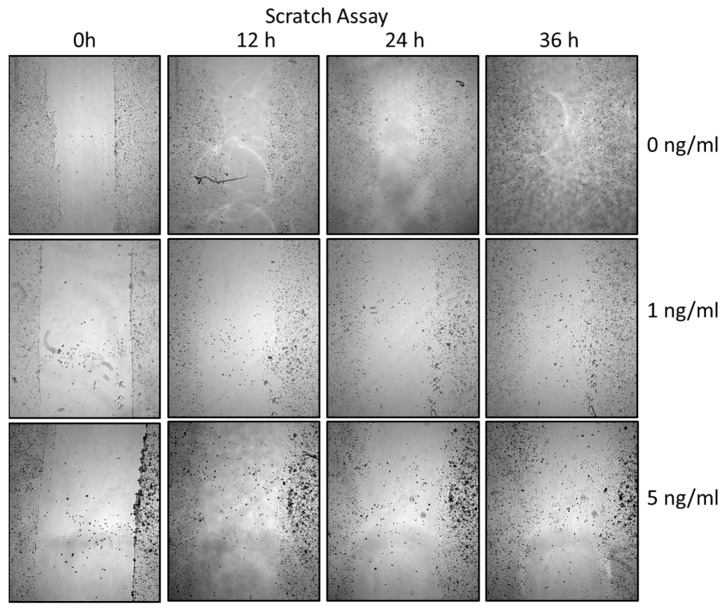
Illustration of the scratch assay performed to verify migration characteristics for the PCI-13 OSCC tumor cell line with (1 and 5 ng/mL) and without (0 ng/mL, control) sclerostin treatment. Magnification ×4.

**Figure 7 cells-13-00137-f007:**
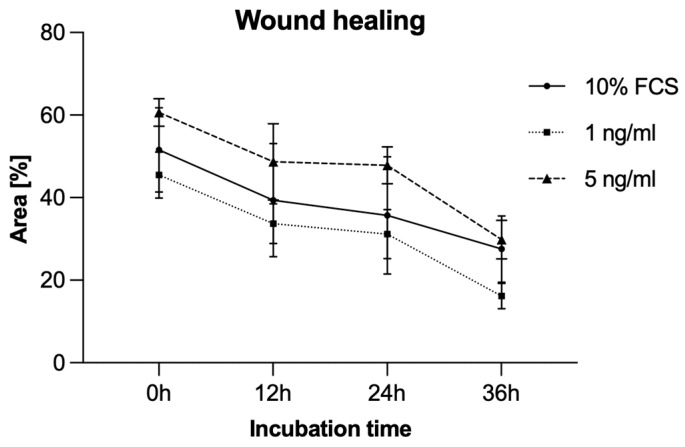
Illustration of the scratch assay to visualize the migration rate of PCI-13 tumor cells under sclerostin treatment at different concentrations (1 and 5 ng/mL) over different time periods (0 h, 12 h, 24 h, 36 h). No significant effect of sclerostin was observed in the scratch assay.

**Figure 8 cells-13-00137-f008:**
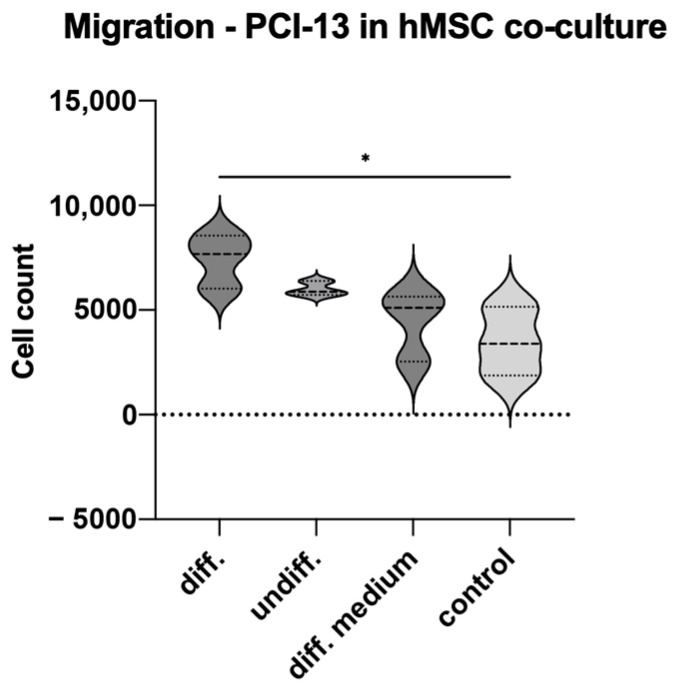
Illustration of the migration rate of PCI-13 tumor cells in co-culture with osteogenically differentiated and undifferentiated hMSCs, as well as in monoculture treated with differentiation medium. Monocultured PCI-13 cells in normal cell culture medium serve as control. Statistical test: one-way ANOVA with Tukey’s tests for post hoc comparisons; * *p*-values between 0.01 and 0.05 (significant). A significant influence of osteogenically differentiated hMSCs on the migration rate of PCI-13 tumor cells was demonstrated.

**Figure 9 cells-13-00137-f009:**
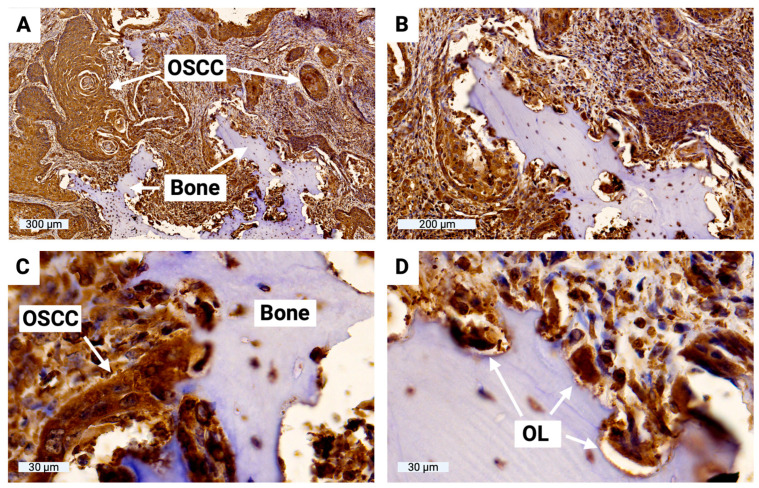
Histological illustration of the OSCC–jawbone interface with additional sclerostin immunohistochemistry. (**A**) Overview of the boundary between the local jawbone and the surrounding tumor cell clusters of OSCC; (**B**) enlarged view; (**C**) highly magnified image of direct tumor cell–bone contact with osteolysis lacunae; (**D**) visualization of multinucleated osteoclasts within osteolysis lacunae (OL). Significant sclerostin expression was found in both OSCC tumor cells and bone-associated cells at the OSCC–jawbone interface.

**Table 1 cells-13-00137-t001:** Primer sequences.

Gene	Sense Sequence	Antisense Sequence
*Osteopontin*	CATATGATGGCCGAGGTGATAG	AGGTGATGTCCTCGTCTGTA
*RUNX2*	CATCACTGTCCTTTGGGAGTAG	ATGTCAAAGGCTGTCTGTAGG
*COL1A1*	CCTGTCTGCTTCCTGTAAACTC	GTTCAGTTTGGGTTGCTTGTC
*ALPL*	GGAGTATGAGAGTGACGAGAAAG	GAAGTGGGAGTGCTTGTATCT
*Osteocalcin*	AAATAGCCCTGGCAGATTCC	CAGCCTCCAGCACTGTTTAT
*Osterix*	GCAAAGCAGGCACAAAGAAG	CAGGTGAAAGGAGCCCATTAG
*Sclerostin*	GGTGAGAGAGAGAGAGAGAAAGA	CTGTCAGAAGAGAGCATCACAA
*GADPH*	GGTGTGAACCATGAGAAGTATGA	GAGTCCTTCCACGATACCAAAG

**Table 2 cells-13-00137-t002:** Clinical patient characteristics and H-score values derived from semiquantitative immunohistochemical evaluation of sclerostin expression at the OSCC–jawbone interface. pT indicates the extent of primary OSCC (pT2: tumor extent ≤2 cm with depth of invasion >5 mm and ≤10 mm; or tumor extent >2 cm and ≤4 cm with depth of invasion ≤10 mm; pT3: tumor extent >2 cm and ≤4 cm with depth of invasion >10 mm; or tumor extension >4 cm with depth of invasion ≤10 mm; pT4a: tumor extension >4 cm with depth of invasion >10 mm; or tumor invades adjacent structures (e.g., through the cortical bone of the mandible or maxilla). pN indicates cervical lymph node metastasis (pN0: no lymph node metastasis; pN2b: metastasis in multiple ipsilateral nodes, none >6 cm in largest dimension, and no extranodal extension. pN2c: metastasis in bilateral or contralateral lymph nodes, none >6 cm in largest dimension, and no extranodal extension; pN3b: metastasis in one or more nodes and clinically evident extranodal extension). pM indicates distant metastases (pM0: no distant metastases). H-score values range from 0 to 300, with 0 indicating that no cells are positive and 300 indicating that all cells are strongly positive.

Sex	Age	Localization	pT	pN	pM	Grading	AJCC Stage	H-Score
Male	65	Gum	4a	0	0	1	IVA	61
Female	54	Gum	4a	2b	0	2	IVA	93
Male	67	Gum	4a	0	0	2	IVA	120
Male	59	Cheek mucosa	4a	3b	0	3	IVB	112
Female	65	Gum	4a	0	0	2	IVA	109
Female	81	Gum	4a	0	0	2	IVA	61
Female	72	Floor of mouth	4a	2c	0	2	IVA	105
Female	75	Gum	3	2b	0	3	IVA	92
Male	63	Gum	4a	2b	0	2	IVA	84
Female	86	Gum	4a	0	0	2	IVA	107
Male	77	Gum	2	0	0	2	II	60
Male	76	Palate	4a	0	0	2	IVA	81
Male	81	Gum	3	2b	0	2	IVA	94
Female	48	Palate	4a	2b	0	2	IVA	100
Male	56	Floor of mouth	4a	2c	0	2	IVA	127

**Table 3 cells-13-00137-t003:** Immunohistochemical staining protocol.

Antigen	Antibody	Pretreatment	Detection Method	Source
Sclerostin	Mouse, monoclonal, clone AbD09097_h/mIgG 2a, 1:1200	HIER (pH 9)	Dako EnVision FLEX	BioRad, Hercules, CA, USA (HCA230Z)

## Data Availability

All data can be obtained from the corresponding author.

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
