# Peer review of "Sclerostin Alters Tumor Cell Characteristics of Oral Squamous Cell Carcinoma and May Be a Key Player in Local Bone Invasion"

_cells, 2024, doi:10.3390/cells13020137_

Round 1

Reviewer 1 Report

Comments and Suggestions for Authors The problem statement is clearly formulated. The introduction is coherent and leads to the research question.
Please revise the following points:
What does the abbreviation "Wnt" mean? This should be explained and why it is important or problematic if this signalling pathway is interrupted. The same applies to Runx2 and CBFb, TGF-ß and SOST. BMP was also explained at the beginning, although this is very familiar. Regarding qRT-PCR analysis: which device was used? How many cycles were run? How was the delta-delta Ct determined? Were there any corrections based on NTCs and positive controls? Regarding the statistical figures: I am aware that Graph-Pad-Prism always produces these bar charts with standard errors-of-the-mean, but they are unscientific. This is a waste of colour, as it basically shows nothing more than the mean value (a number). To recognise the essential characteristics of data, box plots are recommended. Therefore, please replace all bar charts with box plots. If you want to be en vogue, you could also show violin plots instead. Regarding the statistics: an ANOVA was used, which is correct, but how were the pairwise comparisons carried out? This requires a correction for multiple testing, e.g. Bonferroni."

Author Response

  1. The problem statement is clearly formulated. The introduction is coherent and leads to the research question. Please revise the following points: What does the abbreviation "Wnt" mean? This should be explained and why it is important or problematic if this signaling pathway is interrupted. The same applies to Runx2 and CBFb, TGF-ß and SOST. BMP was also explained at the beginning, although this is very familiar. Regarding qRT-PCR analysis: which device was used? How many cycles were run? How was the delta-delta Ct determined? Were there any corrections based on NTCs and positive controls? Regarding the statistical figures: I am aware that Graph-Pad-Prism always produces these bar charts with standard errors-of-the-mean, but they are unscientific. This is a waste of color, as it basically shows nothing more than the mean value (a number). To recognize the essential characteristics of data, box plots are recommended. Therefore, please replace all bar charts with box plots. If you want to be en vogue, you could also show violin plots instead. Regarding the statistics: an ANOVA was used, which is correct, but how were the pairwise comparisons carried out? This requires a correction for multiple testing, e.g. Bonferroni."

Authors' Response: We would like to thank reviewer 1 for critically evaluating our study, praising our work, and providing useful comments that we believe have improved the quality of the manuscript. We have substantially revised all parts of the manuscript and also expanded the background of the Wnt signaling pathway in the Introduction section of the manuscript to clarify the importance of sclerostin in inhibiting this pathway. We have also added short explanations for the abbreviations you provided and provided additional details in the Materials and Methods section for the qRT-PCR experiments. We have changed the figures to violin blots according to your suggestions. The statistical test used was one-way ANOVA with Tukey's post hoc test. We have added references to the manuscript where appropriate. All changes can be found in the revised version of the manuscript.

Reviewer 2 Report

Comments and Suggestions for Authors

Title: "Sclerostin Alters Tumor Cell Characteristics of Oral Squamous Cell Carcinoma and May Be a Key Player in Local Bone Invasion"

Date 17 Dec 2023

1. Abstract

·       The study employs a mix of in vitro experiments and clinical case evaluations, indicating a robust approach.

·       The use of qRT-PCR, immunohistochemistry, and assays for proliferation, migration, and invasion adds depth to the methodology.

·       Both in vitro and clinical data are mentioned, providing a good balance between experimental and real-world evidence.

·       The findings about sclerostin's role in OSCC and bone invasion are clearly stated, but it might be beneficial to briefly mention any limitations or scope for future research in the abstract.

·       The abstract is written in clear, technical language appropriate for the target audience.

·       Usage of specific terms (like qRT-PCR, rh-sclerostin, hMSCs) is appropriate for a scientific audience, though a brief explanation could aid wider accessibility.

·       The abstract concisely summarizes the study's aim, methods, key findings, and implications, which is ideal.

·       The chosen keywords are highly relevant to the study's focus and will aid in its discoverability.

Suggestions for Improvement

·       Consider adding a layman's summary or a more accessible explanation of the technical terms and procedures for a broader audience.

·       The abstract effectively outlines the study, indicating thorough research and methodological rigor. It strikes a balance between technical detail and readability, making it suitable for its intended academic audience. The focus on sclerostin's role in OSCC and bone invasion is clearly conveyed, highlighting the novelty and relevance of the research.

2. Introduction

The introduction provides a comprehensive overview of the background and rationale for the research. Here's a critical analysis of various elements of the introduction:

·       The introduction demonstrates a thorough understanding of the current state of research in the field of OSCC and sclerostin's role in bone metabolism and cancer.

·       The cited literature is highly relevant and provides a solid foundation for the study.

·       The citations are up-to-date and encompass a broad range of relevant topics, including the biology of OSCC, bone invasion, and the role of sclerostin in various cancers.

·       The introduction references studies from various related fields, showing a multidisciplinary approach.

·       The citations appear to be used correctly to support key points. Each reference is directly relevant to the statement it supports.

·       The language is clear, technical, and suitable for an academic audience.

·       The grammar is largely correct, aiding in the readability and understanding of the text.

·       The introduction effectively identifies a gap in the current understanding of sclerostin's role in OSCC, particularly in the context of bone invasion.

·       The study's objectives are clear and logically flow from the background information provided. The rationale for focusing on sclerostin in the context of OSCC is well-articulated.

Suggestions for Improvement

1. Briefly discussing the potential broader clinical or therapeutic implications of this research could enhance the introduction.

2. While comprehensive, the introduction could be made more concise without losing essential information, which might improve its impact.

3. More emphasis on the novelty of this research, particularly in the context of OSCC, could be beneficial.

Overall Reflection

The introduction is well-crafted, demonstrating a deep understanding of the subject matter and effectively setting the stage for the research. It articulates a clear rationale for the study, identifies a specific knowledge gap, and lays a solid foundation for the ensuing research. With minor enhancements, it could be even more impactful.

3. Methodology

·       The "Materials and Methods" section of the study on the influence of sclerostin on the bone invasiveness of Oral Squamous Cell Carcinoma (OSCC) outlines a detailed and methodologically sound approach.

·       The use of three distinct approaches (treatment of tumor cells, co-cultivation, and immunohistochemical analysis) demonstrates a comprehensive and multi-faceted methodological framework.

·       The specifics of cell culture conditions, assays, and staining protocols indicate a high level of rigor and thoroughness.

·       Each method directly contributes to answering the research question, showing relevance and appropriateness.

·       Referencing established protocols and modifications (e.g., qRT-PCR analysis, migration and invasion assays) enhances the relevance of the methods used.

·       The section is detailed, but some parts might be too technical for readers not familiar with laboratory procedures.

·       The subdivision into numbered sub-sections aids in navigating the methodology.

·       The methods represent current practices in cellular and molecular biology research.

·       The language is appropriately technical, matching the expected academic level.

·       The section is well-written with minimal grammatical or syntactical errors.

Suggestions for Improvement

·       Consider adding brief explanations of technical terms and protocols for readers outside the specific field.

·       Including figures or diagrams illustrating complex procedures (like the transwell setup for migration assays) could enhance understanding.

·       More explicit justification for the sample sizes used in each experiment could strengthen the methodological rigor.

·       Emphasize the control conditions in each experiment for clarity.

Here are some suggestions for enhancement:

·       Expand on the descriptions of key methods and techniques, especially for complex procedures or those unique to the study.

·       Include details about the rationale behind choosing specific methods or experimental conditions.

·       Provide a rationale for the chosen sample sizes. Explain how these sizes are sufficient to achieve statistical significance or reliability of results.

·       Clearly define and describe the control conditions for each experiment. Explain why these controls are appropriate and how they contribute to the validity of the results.

·       Specify the number of technical and biological replicates used in each experiment. Explain how this ensures the reliability and reproducibility of the results.

·       Mention any steps taken for standardization and calibration of equipment and procedures. This is crucial for experiments involving sensitive measurements.

·       Outline the statistical methods used for data analysis in greater detail. Explain why these methods are suitable for the data and research questions.

·       Include information about any software or tools used for statistical analysis.

·       Discuss potential sources of bias or error in the methods and how they were mitigated.

·       Mention any troubleshooting steps taken during the experiments.

·       Include diagrams, flowcharts, or other visual aids to illustrate complex experimental setups or procedures.

·       Simplify technical jargon where possible or provide explanations for complex terms and concepts. This makes the study more accessible to readers who may not be specialists in the field.

·       If there is extensive methodological detail, consider including it in supplementary materials and reference this in the methodology section.

These improvements can enhance the transparency, reproducibility, and overall quality of the methodology, making the study more robust and credible in the scientific community.

Overall Reflection

The methodology section is comprehensive, detailed, and methodologically sound, indicating a high level of scientific rigor. It effectively communicates the processes and protocols used in the study, although some parts may be too technical for a general audience. Including visual aids and simplifying some technical details could make the section more accessible. Overall, it forms a strong foundation for the experimental aspects of the study.

4. Results

The "Results" section is well-structured and provides a comprehensive overview of the experimental findings.

·       The presentation of data from various experiments, including qRT-PCR, Boyden chamber assay, scratch assay, and co-culture experiments, demonstrates methodological rigor.

·       The use of one-way ANOVA for statistical testing is appropriate for the data presented, and the significance levels are clearly indicated.

·       The results directly address the study’s objectives, showing the effect of sclerostin on OSCC tumor cell lines in various contexts.

·       The inclusion of data from clinical samples (OSCC jawbone interface) enhances the clinical relevance of the findings.

·       The data is presented in a clear and structured manner, with figures complementing the text.

·       While the section is rich in data, it remains accessible to readers with a basic understanding of the techniques used.

·       The chosen statistical tests are suitable for the type of data and experimental design.

·       The results of the statistical tests are reported in a clear and understandable way, with significance levels appropriately indicated.

·       The results are presented with a focus on data rather than interpretation, which is appropriate for a results section.

·       The findings correlate well with the study's hypotheses and provide a solid basis for discussion.

·       The language is technical and appropriate for a scientific audience.

·       The section is well-written with minimal grammatical or syntactical errors.

Suggestions for Improvement

1. While the figures are helpful, additional graphical representations (like bar graphs or scatter plots) could further enhance data visualization.

2. Some figure descriptions could be more detailed to improve understanding without needing to refer back to the main text.

3. More explicit comparisons between different cell lines and conditions would be beneficial, especially for readers less familiar with the specifics of the field.

4. Ensure consistency in the presentation of data across different figures and tables for easier comparison.

Overall Reflection

The results section is detailed, methodologically sound, and well-presented, effectively conveying the outcomes of the various experiments conducted. It provides a strong foundation for the discussion and conclusion sections of the paper, linking back to the study's objectives and hypotheses. The use of appropriate statistical tests and clear presentation of findings contributes to the overall strength of the study.

To enhance the "Results" section, consider the following suggestions for improvement:

1. Incorporate additional graphical representations such as bar graphs, scatter plots, or heat maps to present complex data in a more digestible and visually appealing manner.

 - Use color coding or labeling to distinguish between different experimental conditions or cell lines.

2. Provide more comprehensive figure captions that explain what each part of the figure represents. This will help readers understand the figures without constantly referring back to the text.

   - Include brief methodological notes in figure captions where necessary, especially for complex procedures.

3. Offer more explicit comparisons between different cell lines, treatments, and conditions within the text. This can help highlight key findings and differences more clearly.

   - Where applicable, briefly relate the findings to existing literature to provide context and show how the results fit into the broader field.

4. Ensure consistency in data presentation across figures and tables. For example, use the same units of measurement and similar scales where appropriate.

   - Clarify any technical terms or abbreviations the first time they are used in the results section, even if they were previously defined in the methods section.

5. Use descriptive subsection headings within the results section to guide the reader through different aspects of the findings.

6. When presenting results from specific methods (e.g., qRT-PCR or Boyden chamber assay), briefly remind the reader of the key aspects of these methods or refer them to the relevant subsection in the Methods section.

7. Consider adding brief summary statements at the end of each subsection, especially for sections with dense or complex data. These summaries can help reinforce key findings.

8.If applicable, include and discuss negative or null results. This can provide a more comprehensive understanding of the research and contribute to the field's knowledge base.

9.Mention any quality checks or reproducibility measures taken during the experiments, such as technical replicates or controls.

10.Although the primary audience may be specialists, consider adding explanations or annotations that make the results accessible to a broader scientific audience.

Implementing these suggestions can enhance the clarity, comprehensiveness, and impact of the results section, thereby strengthening the overall presentation of the research findings.

5. Discussion

·       The discussion integrates findings from the study with existing literature, showing a deep understanding of the topic.

·       The interpretation of results is consistent with the data presented in the results section, maintaining scientific rigor.

·       Discuss the limitations of the methodologies and how they might impact the interpretation of results

·       The section covers a broad range of relevant literature, indicating thorough research.

·       Citations are relevant and appear to be up-to-date, reflecting current knowledge in the field.

·       Citations are used appropriately to support key points and arguments.

·       Citations are correctly used and appear to be accurate based on the context provided.

·       There is a good mix of primary and secondary sources, contributing to a balanced perspective.

·       The discussion reflects the current understanding of the field and integrates recent findings.

·       The language is clear and professional, suitable for an academic audience.

·       The grammar and syntax are generally good, with minimal errors.

·       The discussion seems original in its interpretation and synthesis of data, with no obvious signs of plagiarism.

·       The discussion effectively identifies gaps in current knowledge, particularly regarding the mechanisms of OSCC bone invasion.

Conclusion

·       The conclusions are clear, concise, and directly related to the study's objectives.

·       The study's objectives are addressed, and the conclusions drawn are well-supported by the results.

Suggestions for Improvement

1. Discuss the broader implications of the findings for OSCC treatment and patient management.

2. More explicitly address the limitations of the study and suggest specific directions for future research.

3. Delve deeper into the potential mechanisms by which sclerostin influences tumor cell behavior in OSCC, based on the findings and existing literature.

4. Discuss the potential therapeutic implications of targeting sclerostin in OSCC, considering current treatments and future possibilities.

5. If applicable, discuss the statistical limitations or considerations in interpreting the results.

6. Incorporate perspectives from related fields (e.g., molecular biology, clinical oncology) to enrich the discussion.

7. Consider the impact of these findings on patient outcomes, quality of life, or clinical decision-making.

8. Compare the results more directly with key studies in the field, highlighting similarities and differences.

The "Discussion" section is well-constructed, effectively synthesizes research findings, and provides meaningful insights into the role of sclerostin in OSCC. Addressing the suggested improvements could further enhance its depth and impact.

6. Overall assessment and suitability for publication

·       The study employs a comprehensive methodology, including in vitro experiments and clinical sample analysis, demonstrating scientific rigor.

·       The focus on sclerostin in OSCC represents a novel aspect of cancer research, adding new insights to the field.

·       The study's exploration of sclerostin's role in OSCC bone invasion has significant clinical implications, particularly for understanding tumor progression and potential therapeutic targets.

·       The study contributes valuable knowledge to the field of cancer biology and bone metastasis, which aligns well with the scope of "Cells."

·       The manuscript is well-structured, with clear sections for introduction, methodology, results, discussion, and conclusions. The language and grammar are appropriate for an academic audience.

·       The results are presented with clarity, supported by figures and statistical analysis, making them accessible and understandable.

·       The study demonstrates a thorough understanding of the current literature, with relevant and up-to-date citations.

·       The discussion provides a thoughtful interpretation of the results, placing them in the context of existing knowledge and identifying potential implications.

·       The study appears to adhere to ethical standards, particularly in the handling of human samples.

·       The research methodology and data presentation suggest a high level of scientific integrity.

·       Manuscript Strengths: Novelty, scientific rigor, relevance to current clinical challenges in OSCC, and thorough analysis.

·       Enhance the discussion on broader implications, address limitations more explicitly, and possibly expand on the therapeutic potential of the findings.

Author Response

  1. Abstract
    1. Consider adding a layman's summary or a more accessible explanation of the technical terms and procedures for a broader audience.

Authors' Response: The authors would like to thank reviewer 2 for the critical and detailed evaluation of their manuscript. We have tried to incorporate all of your suggestions for improvement in the appropriate sections of the manuscript and will briefly discuss each point below. As you suggested, we have included a simple summary at the beginning of the manuscript to make the content more understandable to a broader audience.

  1. Introduction
    1. Briefly discussing the potential broader clinical or therapeutic implications of this research could enhance the introduction.
    2. While comprehensive, the introduction could be made more concise without losing essential information, which might improve its impact.
    3. More emphasis on the novelty of this research, particularly in the context of OSCC, could be beneficial.

Authors' Response: We have adapted the Introduction section of the manuscript to include a final reflection on the potential clinical implications of this research. We have also tried to make the Introduction section more concise to improve its impact. At the suggestion of another reviewer, we have added an additional section on the Wnt signaling pathway. We have highlighted the novelty of this research in the context of bone invasion in OSCC at the end of the Introduction section.

  1. Material and Methods
    1. Expand on the descriptions of key methods and techniques, especially for complex procedures or those unique to the study.
    2. Include details about the rationale behind choosing specific methods or experimental conditions.
    3. Provide a rationale for the chosen sample sizes. Explain how these sizes are sufficient to achieve statistical significance or reliability of results.
    4. Clearly define and describe the control conditions for each experiment. Explain why these controls are appropriate and how they contribute to the validity of the results.
    5. Specify the number of technical and biological replicates used in each experiment. Explain how this ensures the reliability and reproducibility of the results.
    6. Mention any steps taken for standardization and calibration of equipment and procedures. This is crucial for experiments involving sensitive measurements.
    7. Outline the statistical methods used for data analysis in greater detail. Explain why these methods are suitable for the data and research questions.
    8. Include information about any software or tools used for statistical analysis.
    9. Discuss potential sources of bias or error in the methods and how they were mitigated.
    10. Mention any troubleshooting steps taken during the experiments.
    11. Include diagrams, flowcharts, or other visual aids to illustrate complex experimental setups or procedures.
    12. Simplify technical jargon where possible or provide explanations for complex terms and concepts. This makes the study more accessible to readers who may not be specialists in the field.
    13. If there is extensive methodological detail, consider including it in supplementary materials and reference this in the methodology section.

Authors' Response: The authors have tried to address all of your comments when revising the Materials and Methods section of the manuscript. Since some comments were very frequent, it was not entirely clear to us which section needed further improvement. However, in each section of the Materials and Methods section, the method descriptions have been expanded and items such as the controls used or the number of samples used are now explained in more detail. As a result, the authors believe that the Methods section is now much easier for the reader to understand.

  1. Results
    1. While the figures are helpful, additional graphical representations (like bar graphs or scatter plots) could further enhance data visualization.
    2. Some figure descriptions could be more detailed to improve understanding without needing to refer back to the main text.
    3. More explicit comparisons between different cell lines and conditions would be beneficial, especially for readers less familiar with the specifics of the field.
    4. Ensure consistency in the presentation of data across different figures and tables for easier comparison.

Authors' Response: The authors thank the reviewer for the useful suggestions. At the suggestion of another reviewer, we have changed the figures to violin blots. We have also revised the figure descriptions accordingly to improve understanding, and have tried to make the comparison between cell lines and conditions as explicit as possible to make it easier for the reader to understand. We have evaluated and, where necessary, revised all figures and tables to ensure comparability. All changes are included in the revised version of the manuscript.

  1. Discussion
    1. Discuss the broader implications of the findings for OSCC treatment and patient management.
    2. More explicitly address the limitations of the study and suggest specific directions for future research.
    3. Delve deeper into the potential mechanisms by which sclerostin influences tumor cell behavior in OSCC, based on the findings and existing literature.
    4. Discuss the potential therapeutic implications of targeting sclerostin in OSCC, considering current treatments and future possibilities.
    5. If applicable, discuss the statistical limitations or considerations in interpreting the results.
    6. Incorporate perspectives from related fields (e.g., molecular biology, clinical oncology) to enrich the discussion.
    7. Consider the impact of these findings on patient outcomes, quality of life, or clinical decision-making.
    8. Compare the results more directly with key studies in the field, highlighting similarities and differences.

Authors' Response: We have substantially revised the Discussion section to include a more detailed discussion of the potential benefits of this research for patient care and the potential for drug-based sclerostin inhibition. In addition, we have outlined the limitations of the study and provided an outlook for further investigation. Although little is known about the involvement of sclerostin in tumor bone invasion and the altered properties of tumor cells, we have shed light on the possible molecular mechanisms. Please see the modified version of the discussion in the revised version of the manuscript.

Reviewer 3 Report

Comments and Suggestions for Authors

Detailed comments can be found in the attached file.

Author Response

  1. Line 52-53. It is not entirely clear what authors meant. Can it be further elaborated?

Authors' Response: The authors would like to thank reviewer 3 for critically evaluating our study and providing useful comments, which we believe have improved the quality of the manuscript. Below, we attempt to address each of the points raised to the best of our ability. We have revised the unclear sentence you mention (line 52-53) to clarify the effect of sclerostin on BMPs and refer to the revised version of the manuscript.

  1. Some minor editing would improve the manuscript, for instance: Include what HMSCs stands for (line117).
    1. Replace “elsewhere” in line 156.
    2. Include what DAB stands for (line 226).
    3. Include in the caption of table 1. the meaning of pT, pN and pM should be included.
    4. Figure 4. X-axis legend is trunked.
    5. “asterix”(line 258).

Authors' Response: We have implemented all of your editing suggestions for improvement in the revised version of the manuscript.

  1. Section 2.6. authors refer that the scratch assay experiment was performed over a period of 36 hours, however section 3.5. states that the evaluation period extends up to 60 hours. Can the authors clarify? Additionally, a figure similar to Figure 2. should be presented in the results section instead.

Authors' Response: Indeed, in the scratch assay, cells were generally observed for 60h. However, the scratch could not be sufficiently analyzed after 36h. Therefore, the authors decided to reduce the time points for analysis to 36 h. To avoid further confusion, we have deleted the additional time points in the corresponding section. In addition, the authors have moved Figure 2 to the Results section.

  1. Can the authors provide some details about which and how cell/structures were annotated and how many images were used to do annotations prior to automated analysis? How was the threshold to subclassify the tumor cells established? Authors emphasize the role of sclerostin on osteoclastic activity and its role on tumoral invasion (discussion section). Did the authors consider quantify the number of multinucleated cells/osteolytic lesions within the ROIs and correlate it with the H-score? It would provide additional information on tumoral activity.

Authors' Response: We would like to thank you for this useful comment and will try to explain the automatic analysis with the QuPath software in the following. In each clinical case, three different regions of interest (ROIs) at the OSCC jawbone interface were digitally defined as a square. Each ROI was approximately 1 cm2 in size. Spot vector and background data were used to improve spot separation in the software through color deconvolution. Automated cell detection was used to identify all cells in each ROI based on cell and membrane staining. This procedure additionally estimates the full extent of each cell based on a restricted extent of the nuclear region and calculates up to 33 measurements of intensity and morphology, including nuclear area, circularity, staining intensity for hematoxylin and DAB, and nuclear to cell area ratio. Since not all of these measurements are expected to provide independent or useful information for cell classification, a subset of 16 measurements was empirically selected and expanded for each cell by measuring the local density of the cells and calculating a Gaussian-weighted sum of the corresponding measurements within neighboring cells using the QuPath add smoothed features command. A two-way random tree classifier was trained in the software to distinguish OSCC tumor cells from other cell or tissue types. The software's default intensity thresholds were used to further subdivide tumor cells with negative, weak, moderate, or strong positive staining based on mean optical DAB densities. For each ROI, an H-score was calculated by adding 3x% strongly stained tumor cells, 2x% moderately stained tumor cells, and 1x% weakly stained tumor cells, resulting in scores ranging from 0 (all tumor cells negative) to 300 (all tumor cells strongly positive). Means of the three different ROIs were calculated for statistical analysis. We have included this detailed explanation of automatic analysis in the manuscript.

As you indicated, we were able to detect multinucleated osteoclasts in osteolytic lesions at the jawbone interface of OSCC (Figure 9D), but quantification and correlation with sclerostin expression was not meaningful due to the small number of cases. The extent of surgical intervention currently limits the number of evaluable histologic specimens, and the goal of the presented histologic evaluation was to translate the in vitro results into clinical cases to demonstrate substantial sclerostin expression at the OSCC jawbone interface. However, we plan to comprehensively characterize the influence of sclerostin on the bone invasion of OSCC.

  1. Section 3.1. why are sclerostin relative expression values not calculated in relation to the housekeeping gene (similarly to section 3.2.)?

Authors' Response: We thank the reviewer for this useful comment. The graph has been replaced with a new graph showing the delta CT values of SOST and GAPDH.

  1. Section 2.4. states (line 175) that cell counting was performed using Aivis 4D version 3.2, nonetheless section 3.4. states that nucleus were counted using Fiji image J (line 298). Can the authors clarify?

Authors' Response: We thank the reviewer for the useful comment. The typing error in line 175 has been corrected. Cell counting with Fiji ImageJ in section 3.4 is an error. Cell counting for the analysis of migration and invasion of PCI-13 cells was performed with Arivis 4D. 

  1. Did the authors find any correlations between the tumor staging and the calculated H-score (section2.7.)? If so, that should be acknowledged and further discussed on the discussion section.

Authors' Response: In this study, the authors did not find a statistically significant correlation between the H-score values of sclerostin expression at the jawbone margin of OSCC and the factors of the TNM classification system. However, the aim of the present histologic evaluation was not to demonstrate such a correlation, as this would require a larger number of cases and is currently limited by the complexity of the surgical procedures, but rather to translate the in vitro results into quantified values in selected clinical cases. Nevertheless, in a first pilot study, we show a significant prognostic impact of sclerostin expression in OSCC tumor cells on the disease-free survival (DFS) of patients, which for the first time alerted us to the potential effect of sclerostin in OSCC bone invasion.

Schminke, B., Shomroni, O., Salinas, G., Bremmer, F., Kauffmann, P., Schliephake, H., ... & Brockmeyer, P. (2023). Prognostic factor identification by screening changes in differentially expressed genes in oral squamous cell carcinoma. Oral Diseases, 29(1), 116-127.

Round 2

Reviewer 2 Report

Comments and Suggestions for Authors

Here are my observations on the revised manuscript

Abstract:

·       Adequate

·       All suggested changes were done

Top of Form

Introduction

·       Adequate

·       Significantly improved

·       All suggested changes were done

Material and Methods:

·       Adequate

·       Significantly improved

·       All suggested changes were incorporated in the section

Results:

·       The results section improved

·       Adequate

·       All suggested changes were incorporated in the section

Top of Form

Discussion

·       The discussion section provides an overview of the study findings and compares them to previous research.

·       The results section improved significantly

·      All suggested changes were incorporated in the section

The manuscript now provides a more comprehensive and informative interpretation of the findings, and provide meaningful insights for clinical practice and future research.

Top of Form

Bottom of Form

Author Response

The manuscript now provides a more comprehensive and informative interpretation of the findings, and provide meaningful insights for clinical practice and future research.

Authors' Response: The authors would like to thank reviewer 2 for the important suggestions for improvement and praise of their work.

Reviewer 3 Report

Comments and Suggestions for Authors

The reviewer deeply appreciate the author´s response. Further suggestions concerning the revised version are the following:

1. Figure 7. still presents quantitative data, regarding the scratch assay, for a 60 hour period, despite the results are consdired till the 36 hours period. The same applies to the text in line 403.

2. Revision of the text in lines 281-295 is strongly advised. Although the reviewer understands that the method is based in prior work, the resemblance with the original text of Bankhead et al is uncanny. The reviewer is certain that the authors did not mean to it, so in order to avoid any reader´s missassumptions, revision is recommended.

Author Response

Figure 7. still presents quantitative data, regarding the scratch assay, for a 60 hour period, despite the results are consdired till the 36 hours period. The same applies to the text in line 403.

Authors' Response: The authors would like to thank reviewer 3 for pointing out this error. We have corrected Figure 7 and the corresponding text.

Revision of the text in lines 281-295 is strongly advised. Although the reviewer understands that the method is based in prior work, the resemblance with the original text of Bankhead et al is uncanny. The reviewer is certain that the authors did not mean to it, so in order to avoid any reader´s missassumptions, revision is recommended.

Authors' Response: The authors would like to thank reviewer 3 for this useful comment. We have revised the text according to your comment to avoid any misunderstanding for the reader.